# Dynamic Brain Connectivity in Resting-State FMRI Using Spectral ICA and Graph Approach: Application to Healthy Controls and Multiple Sclerosis

**DOI:** 10.3390/diagnostics12092263

**Published:** 2022-09-19

**Authors:** Amir Hosein Riazi, Hossein Rabbani, Rahele Kafieh

**Affiliations:** 1Medical Image and Signal Processing Research Center, School of Advanced Technologies in Medicine, Isfahan University of Medical Sciences, Isfahan 8174673461, Iran; 2Department of Engineering, Durham University, South Road, Durham DH1 3LE, UK

**Keywords:** functional MRI, brain connectivity, multiple sclerosis, ICA

## Abstract

Multiple sclerosis (MS) is a neuroinflammatory disease that involves structural and functional damage to the brain. It changes the functional connectivity of the brain between and within networks. Resting-state functional magnetic resonance imaging (fMRI) enables us to measure functional correlation and independence between different brain regions. In recent years, statistical methods, including independent component analysis (ICA) and graph-based analysis, have been widely used in fMRI studies. Furthermore, topological properties of the brain have been appeared as significant features of neuroscience studies. Most studies are focused on graph analysis and ICA methods, rather than considering spectral approaches. Here, we developed a new framework to measure brain connectivity (in static and dynamic formats) and incorporate it to study fMRI data from MS patients and healthy controls (HCs). For this purpose, a spectral ICA method is proposed to extract the nodes of the brain graph. Spectral ICA extracts more reliable components and decreases the processing time in calculation of the static brain connectivity. Compared to Infomax ICA, dynamic range and low-frequency to high-frequency power ratio (fALFF) show better results using the proposed ICA. It is also helpful in selection of the states for dynamic connectivity. Furthermore, the dynamic connectivity-based extracted components from spectral ICA are estimated using a mutual information method and based on correlation of sliding time-windowed on selected IC time courses. First-level and second-level connectivity states are calculated using correlations of connectivity strength between graph nodes (spectral ICA components). Finally, static and dynamic connectivity are analyzed based on correlation nodes percolated by an anatomical automatic labeling (AAL) atlas. Despite static and dynamic connectivity results of AAL correlations not showing any significant changes between MS and HC, our results based on spectral ICA in static and dynamic connectivity showed significantly decreased connectivity in MS patients in the anterior cingulate cortex, whereas it was significantly weaker in the core but stronger at the periphery of the posterior cingulate cortex.

## 1. Introduction

Multiple sclerosis (MS) is a central nervous system (CNS) disease that is characterized by multiple lesions occurring mostly in the white matter. As a result, structural and functional connectivity between various areas of the CNS is changed. Functional magnetic resonance imaging (fMRI) in resting state and tasks shows large functional changes in MS patients. The study of resting-state functional connectivity in MS is primarily aimed at understanding changes in the innate functional map of the brain and their role in disease progression and clinical disorders. Resting-state fMRI can be used to recognize distinct regions of the brain that configure specific resting-state networks [1]. Unlike task fMRI, resting-state fMRI is not affected by designed task performance, which may differ from healthy individuals, especially in patients with clinical disabilities.

Axonal injury is a dangerous and critical process in many brain diseases. Understanding the processes associated with such injury is critical for resolving the causes of brain disorders. Table 1 summarizes studies on MS using resting-state functional MRI (rs-fMRI), task-based fMRI, and other MRI modalities. The main disadvantages of the methods listed in Table 1 are unidentified stage of the disease, unclassified background of MS studies, unmatched age and sex between patients and healthy controls (HCs), and dissimilar tasks among different methods, which make comparisons impossible.

Today, hidden information in structural and functional connections between different regions of the brain is reported to be important for diagnosis of mental illnesses at different stages of the disease. In this regard, MRI modalities, such as diffusion tensor imaging (DTI) and fMRI, were introduced to derive structural and functional connections. fMRI measures the response of the hemodynamic system (blood-flow change) associated with neural activity in the human brain or spinal cord. Since the magnetic resonance signal is contrasted with the blood oxygen level, the term “blood oxygenation level-dependent” (BOLD) is known in the fMRI community. Available methods for analyzing functional relationships through fMRI are generally divided into two groups: model-based and data-based. Model-based approaches, e.g., cross-correlation analysis (CCA), need prior knowledge and are easy to implement and widely used. Data-driven methods (either based on clustering or data analysis) require no prior knowledge and are thus quite useful for rs-fMRI studies.

Independent vector analysis (IVA) [19,20] is an approach that keeps variability with extending ICA to multiple datasets. IVA-GL algorithm [21] is a combination of IVA using the Laplace density model (IVA-L) with IVA using the Gaussian density model (IVA-G). IVA-GL assumes super-Gaussian distribution for the sources and uses second-order and higher-order statistical dependence among multiple subjects.

In this study, rs-fMRI is used for HCs and MS with similar stage of disease. Furthermore, a multistage, data-driven method is developed to calculate more accurate functional connectivity. A modified independent component analysis (ICA) method is proposed to overcome speed concerns and local minima problems in time-series analysis. Dynamic connectivity is then extracted with the IVA-GL algorithm and more realistic independent components are selected using clustering and filtering methods.

The rest of the paper is organized as follows. In Section 2, our methods, including the proposed ICA method and brain connectivity using graphs, are described. In Section 3, the results are presented, showing the improvements achieved by the proposed method. Finally, in Section 4, the conclusions are presented.

## 2. Material and Methods

### 2.1. Data Acquisition

Overall, 12 RRMS patients (7 female and 5 male) and 12 HCs (4 female and 8 male) were included in this analysis. The MRI scans were performed with a 3 T MRI scanner (Magnetom Tim Trio, Siemens Healthcare, Erlangen, Germany) with a 32-channel receive-only head coil. MRI used a gradient echo (GE)–EPI sequence with the following settings: TR = 3000 ms, TE = 30 ms, flip angle = 90°, field of view (FOV) = 192 × 192 mm^²^, matrix size = 64 × 64, spatial in-plane resolution 3 mm, 49 slices with a slice thickness of 2 mm and an interslice gap of 1 mm, readout bandwidth (BW) = 2232 Hz/pixel. A time series with 200 time points was acquired.

All subjects were from the Department of Neurology, University Medical Center of the Johannes Gutenberg University Mainz and the Department of Neurology, Goethe-University Frankfurt, and underwent a standardized MRI protocol in Mainz [22]. Prior to participating in this study, informed written consent was collected from all individuals, which was approved by the Ethics Committee of the Rhineland-Palatinate State Medical Association. This study was based on the principles set out in the Helsinki Declaration. Human ethics documents were used before any data collection. Overall, 24 documents were approved by the ethics committee. Each patient with MS was evaluated by an experienced neurologist and an Expanded Disability Status Scale (EDSS) score was determined at study entry after one year. Fatigue was classified according to the Fatigue Scale for Motor and Cognitive Functions (FSMC) [23] and was also assessed at study input after one year. All patients had recurrence at least 60 days before enrollment. To ensure a homogeneous clinical set of patients, patients with clinical recurrence during the study period were excluded [22].

### 2.2. Preprocessing

Parallel auto-preprocessing was used to realign the images. The data were then spatially normalized to the Montreal Institute of Neurological Institute (MNI) standard atlas, 3 mm × 3 mm × 3 mm voxels using nonlinear registration (affine + low-frequency direct cosine transform functions) in SPM8 toolbox (http://www.fil.ion.ucl.ac.uk/spm, accessed on 1 March 2021). A full-width Gaussian kernel at half-maximum 5 mm was used to reduce some false-positive correlations in further analysis.

### 2.3. Methods

We used modified spectral ICA for calculating static and dynamic connectivity and better discrimination of MS and HC. The method block diagram is shown in Figure 1 and listed below.
Extraction of ICs (nodes) using proposed ICA methodCalculation of connectivity map for all datasets and all ICs, as well as extracting connectivity maps for MS and HC.Calculation of static and dynamic connectivityCalculation of standard deviation between windows for MS and HCCalculation of different connections between the graph nodes. Most discriminative nodes are temporal gyrus and frontal gyrus. All main nodes listed in Table 2.

To prove better discrimination of the proposed ICA between MS and HC, we used dynamic range and fAlff of extracted ICs, listed in Table 3. For this purpose, statistical measurements were computed for each standard deviation of functional connectivity between the groups.

#### 2.3.1. Conventional Independent Component Analysis (ICA)

ICA is a statistical method to decompose a set of information (observed data) such that the decomposed sources will be maximally independent. ICA is one of the famous fMRI models used to extract spatial or temporal sources of observed data (mostly the spatial components are extracted). In spatial ICA analysis, X is an observation data matrix with size of N × M, where N is the number of time points and M is the number of voxels. As such, M-dimensional random vector denoted by $X={\left[x_{1},x_{2},\ldots ,x_{M}\right]}^{T}$ (observation matrix) should be solved by ICA matrix [24]:(1)X=AS

To acquire N-dimensional source matrix $S={\left[s_{1},s_{2},\ldots ,s_{N}\right]}^{T}$, whose elements refer to independent sources and A as an unknown mixing matrix. Usually, the number of voxels (M) is greater than the number of time points and “A” is usually a full rank matrix. Good approximation of sources is determined using higher than second-order statistics. Most popular ICA algorithms use nonlinear methods to generate higher-order statistics based on maximum-likelihood estimation, maximization of information transfer, mutual information minimization, and maximization of non-Gaussianity.

One of the main concerns in maximization methods is the selection of a proper starting point to solve the problem of local maxima. For instance, Infomax, one of the most popular algorithms, finds the unknown mixing matrix (*A*) by maximizing the sum of the marginal entropies. Considering the orthogonality rules, joint entropy function is invariant to orthogonal transformation (*H*(*AX*) = *H*(*X*)), where entropy of sources $H\left(S\right)=\sum H\left(s_{i}\right)$ is equal to sum of marginal entropies. Accordingly, Infomax finds A by maximizing $H\left(S\right)$:(2)$$\underset{A}{max}\sum_{i=1}^{N}H\left[{\left(A^{-1}X\right)}_{j}\right]$$

Infomax uses gradient methods for global search and optimization steps. However, Infomax (like other optimization algorithms) encounters two disadvantages: first, complicated nonlinear function in marginal entropy function; second, problem of finding local maximum (which is more prone to error in higher dimension) and cannot be solved by repetition.

#### 2.3.2. Spectral ICA

Graph Laplacian is a known method in clustering approaches, dimension reduction, and classification. In the ICA algorithm, elements of a weighted matrix are calculated using observation matrix (X):(3)$$W_{ij}=k\left(∥X^{i}-X^{j}∥{}^{2}/2\varepsilon \right)$$
where ‖.‖ is the Euclidean distance, k is an exponential kernel and ε is bandwidth of the kernel ($k\left(x\right)=e^{-x}$). The normalization matrix (*D*) is then calculated as sum of the weighted matrix elements:(4)$$D_{ii}=\sum_{j=1}^{N}W_{ij}$$

The weighted matrix is then normalized with matrix *D* and the negative definite graph Laplacian matrix (*L*) is defined by:(5)$$L=D^{-1}W-I$$
where I is an N × N identity matrix. In spectral ICA, it is proven that the top few eigenvectors belonging to nondegenerative eigenvalues of L are tightly correlated with independent component of the observation matrix [25]. It has also been shown that spectral ICA is not reliable for extracting more than one or two ICs (with more than 90% correlation to outputs of ICA); however, the correlation value for the next eigenvectors decreases to lower than 60%. Compared with ICA, spectral ICA has the advantage of exemption from wrong initial value selection and local maxima, but is more time-consuming.

#### 2.3.3. Proposed Modified Spectral ICA Method

To implement, first the number of ICs is estimated using the minimum description length (MDL) method (R is estimated number of ICs) [26]. Graph Laplacian matrix is then calculated according to Equation (5) and first R nondegenerative eigenvalues are selected and corresponding eigenvectors are extracted. These eigenvectors are then used as initial ICs for the Infomax algorithm. The mixing matrix of Infomax is calculated based on the observation matrix and the initial components ($A_{initial}=X\cdot S_{spectral}^{-1}$) instead of a random mixing matrix. In our study, a new group ICA method on fMRI data is introduced by designing a modified combination of spectral ICA and Infomax algorithm and formulating a new approach in calculation of brain connectivity, elaborated in section C and D. The Infomax algorithm and some utilities like repetition of ICA with ICASSO were undertaken using the GIFT software package from MATLAB (http://mialab.mrn.org/software/gift/, accessed on 15 April 2021) [27].

#### 2.3.4. Proposed Method for Extraction of Static and Dynamic Brain Connectivity

For each individual, an m×m (where m is number of ICNs selected as 37 in this study) weighted matrix is first constructed that represents a stationary brain connectivity (static connectivity at the time) graph using the entire data time. ICs overlay to brain atlas to know their Brodmann area. The departed ICs based on Brodmann area atlas are known as ICNs. Selected ICs are in contrast to ICs related to physiological activity, movement, or imaging artifacts (ARTs). These components are evaluated based on the expectation that ICNs should reveal peak activations in gray matter, low spatial overlap with known ARTs, and should represent time courses (TCs) that are correlated with low-frequency fluctuation. The connectivity graph approach helps us to find the relationship between the two ICNs. To build the edges in the connectivity graph, the correlation of time courses is calculated for all ICN pairs, where *B* is *n × n* degree matrix of *K.*

The dynamic connectivity analysis of each individual is performed using a time-sliding window (with width of L=22×TR (66 s) in steps of 1×TR sliding), according to the findings of Allen et al. (2014) [28] for establishing a good trade-off between the ability of solving dynamicity and the quality of the connection estimation. Sliding windows are implemented to divide total time points for each subject into smaller datasets. The IVA-GL algorithm is applied on all these smaller datasets for all subjects. Back-reconstructed component maps are acquired for each subject at each time window. The m=37 ICN TCs are divided into time segments by taking the time windows of width L, leading to F−L+1=179 distinct matrixes (where *F* is number of time points). Then, 179 different weighted brain graphs m×m are constructed ($S^{w},w=1,2,3,\ldots ,179$) and graph measurements, including the connectivity strength, clustering coefficient, and global efficiency of $S^{w}$S are calculated by the Brain Connectivity Toolbox [29].

## 3. Results

The proposed modified spectral ICA method takes advantage of both spectral ICA and Infomax algorithms. Spectral ICA extracts independent components quickly, but not accurately; and Infomax is time-consuming and has a local maxima problem (due to random initialization). The proposed method takes mutual advantage of both methodologies: the exact optimization is borrowed from Infomax to yield accurate ICs, and good initialization is taken from spectral ICA to provide a reliable initial step and yielding to fast and trustworthy convergence.

### 3.1. Modified Spectral ICA and Stationary Connectivity

The number of ICNs was extracted using minimum description length (MDL). A spatial map of the ICNs was extracted with the proposed ICA algorithm. The ICA algorithm was repeated 10 times in the ICASSO decomposition reliability estimation and the resulting clusters compacted [30]. Also, by calculating the power spectrum, only the components with low-frequency power spectra were selected [31]. Finally, 37 independent components (ICs) were extracted as intrinsic connectivity networks (ICNs). Selected ICs were in contrast to ICs related to physiological activity, movement, or imaging artifacts (ARTs). These components were evaluated based on the expectation that ICNs should reveal peak activations in gray matter, low spatial overlap with known ARTs, and represent time courses (TC) correlated with low-frequency fluctuation. As demonstrated in Figure 2, ICNs were rearranged based on their regions and their dynamic range. In the power spectra, at frequency of right side of the peak, the difference between the maximum power and minimum power was calculated. In addition, to compare the proposed ICA and Infomax ICA, fALFF (low-frequency to high-frequency power ratio) and dynamic range of extracted ICs were compared (Table 3). The similarity matrix between ICNs, known as stationary functional connectivity for each subject, was computed from the entire scan. Then, the final similarity matrix was computed by averaging the similarity matrix for all subjects in a group. Functional network connectivity correlations were also computed for each dataset and averaged across sessions. In addition, mutual information was computed between components spatially and averaged across the datasets. Stationary functional connectivity of the MS group and healthy group are shown in Figure 3.

### 3.2. Dynamic Connectivity and Graph Properties

Stationary connectivity, as a first-level analysis, was based on 37 identified connectivity states, based on the extracted ICs. Dynamic functional connectivity with 50% sliding windows were calculated to divide total time points for each subject into smaller datasets. Mean and standard deviation of all extracted window in this dataset did not yield meaningful results.

A possible problem is the number of windowed FNC correlations used by the dynamic method for extracting state by reducing the number of FNC correlation windows to a few clusters. This process can be done with different methods, such as k-mean, ICA, and PCA. We used k-mean clustering to extract six different states from all extracted windows. Dynamic connectivity of six different states extracted with k-mean clustering is shown in Figure 4.

Mean values and standard deviation were computed across windows for all datasets and averaged across the subjects (Figure 5). To reduce the number of windowed FNC correlations, ICA analysis with Infomax was repeated on all extracted windows. We extracted six states with this method, more reliable than k-mean (Figure 4). ICA-based extracted states are shown in Figure 6.

For each group, we constructed a dynamic connectivity map. At the first step, a chart of state vs. mean dwell time was determined and is shown in Figure 7. This chart proves that the second state is more reliable than other extracted states. Then, back-reconstruction of the second state for HC and MS groups was done and two-sample *t*-test results run (Figure 8).

As another method to compare ICA-based static and dynamic connectivity, we analyzed the connectivity based on correlation of each pair of the regions in atlas of AAL. Static and dynamic functional connectivity maps of HC and MS using this correlation are shown in Figure 9 and Figure 10.

### 3.3. Assessing Reliability and Stability in ICA for fMRI Data

One of the most important problems in ICA in analysis of fMRI data is the stability and reliability of the extracted independent components. The first source of uncertainty in ICA algorithm is the number of components to be extracted. Minimum description length (MDL) is used for estimating the number of ICs. Another source of uncertainty is using random initialization to find local minima in most of the algorithms.

To overcome the first uncertainties (number of estimated ICs), we used the MDL algorithm in most of the proposed algorithms. However, we also used the R-index, which defines the degree of difference between two samples. Figure 11 shows that the best results were achieved by ICs between 32 to 37.

Regarding the other source of uncertainty (random initialization), we had two strategies to overcome the local minimum problem. The first strategy is using spectral ICA, where the type of the optimization method with graph Laplacian is robust to change in repetitions. The proposed method uses spectral ICA as an input of the ICA algorithm, and it yielded stability of final extracted ICs. In addition, we used ICASSO, which uses repetition and bootstrap, to find the more reliable and stable ICs (Figure 12).

## 4. Discussion

In this study, the characteristics of the dynamic graph were determined by calculating different connections among different brain functions in HCs and MSs in resting-state fMRI data. Brain graph nodes are defined by ICNs detected by group spectral ICA. Dynamic weight brain graph is created using time-window sliding correlation analysis. The results indicated that the dynamic graph metrics discovers higher variance in HC than MS. Median value for connectivity measurements in MS are reduced.

We demonstrated three distinct kinds of comparisons to show the effectiveness of the proposed method. First, results of the proposed method are compared with previous publications with the most similar methods. Second, results of the proposed method are compared with conventional methods using the same dataset. Third, results of the proposed method are compared with medical achievement for MS patients.

The first declaration about validation of results is about connectivity changes. Our results show that reduction in first-level connectivity state is associated with second-level connectivity states. In first-level analysis, most of the changes are happening in third, fourth and ninth ICs, and in second-level analysis, there are a lot of changes in connections between HC and MS. For example, seventh IC connections are completely different in each group. In other word, static connectivity shows only some changes in frontal gyrus, but dynamic connectivity is able to discover the changes in frontal gyrus, cuneus, caudate and inferior parietal lobule.

In comparison with other publications with the most similar methods, dynamic connectivity has more reliable results. A second comparison is on performance of ICA vs a conventional method (correlation with AAL90 atlas). As shown in Figure 9 and Figure 10, the results of correlation could not determine any changes between MS and HC; however, first-level and second-level connectivity using the proposed method is able to successfully extract obvious changes

In blind source separation methods, uncertainty and reliability must be discussed. ICASSO is well known for using bootstrap and multiple random initialization to check stability. In this paper, we apply real ICASSO and ICASSO, and as per Figure 12a, real ICASSO shows concertation and repeatability of the extracted ICs. Figure 12b presents the results of ICASSO with proposed ICA. It is clear that estimated clusters in proposed ICA are more reliable than conventional ICA. It is because of random initialization in ICA. In addition, the number of ICs, as another issue in uncertainty analysis, is addressed by calculating R-index between all ICs after applying ICA. By this method, the similarity between all clusters is checked. As shown in Figure 11, after 32 clusters, R-index is minimized. Finally, we select the number of ICs based on MDL calculation to select and the most valid number of ICs and to decrease ICA uncertainty.

Our findings showed that the characteristics of the dynamic graph change based on the functional connection of the variable brain together in MS, which is a potentially new indicator for this mental illness. This provides a new framework for evaluating dynamic brain graphs in resting-state fMRI data.

Due to the massive quantity and smallness of the neurons, it is challenging to build a complete brain network at the neural level [32,33]. Functional brain connection networks in fMRI data are often based on connections between large areas of the brain on a macroscopic scale. Defining brain network nodes based on parcellation approaches includes the use of predefined anatomical formats such as automatic anatomical labeling (AAL) [34,35,36], randomly generated templates [37,38], and voxel-based divisions [39,40]. Few measures of the topological characteristics of the brain graph may be modified by [32,33]. Previous studies have estimated the network when using atlas-based areas (ROI) as damaged graph nodes [41,42,43]. Furthermore, ROIs certainly do not respect the functional regions of the human brain or regions that reflect individual differences in subjects. In contrast, ICA provides a data-driven approach to build networks by defining brain components as functionally similar nodes [44,45,46,47]. In accordance with previous studies that also defined network nodes using spatial ICA [28,47,48], in this study, the brain connectivity indicates a modular organization within somatomotor, visual, cognitive control, default mode, and auditory regions, as well as anticorrelation between default mode, visual, and auditory regions.

Beyond the static connectivity, we used correlation analysis based on time-sliding windows to obtain dynamic topological criteria of time-varying brain graphs. This is the most common strategy for investigating the brain connectivity dynamics in resting-state fMRI [49,50]. Recent studies used this approach to show dynamics of the brain connectivity. Sakoglou et al. (2010) [51] investigated dynamics of functional network connectivity in schizophrenia. Jones et al. (2012) [52] examined the dynamic connectivity of brain networks in Alzheimer’s disease. Wee et al. (2013) [53] used this approach to detect early mild cognitive impairment (eMCI). However, these studies did not calculate the measurement of the dynamic graph, which numerically analyzes dynamicity function of the whole brain. There are few studies that analyze this type of dynamicity. In comparison with this study, in previous literature, Pearson correlation was used, which is the most classic approach to determine the existence of linear relationships via correlation coefficients, but we used mutual information to test network dependency. We applied a different ICA method that increased accuracy and speed of ICs. We compared both node selection method, atlas-based node selection and data-driven (ICA-based) node selection. Results showed that data-driven node selection was the most accurate method, with more differences between HC and MS. Table 2 shows the significant difference between two groups that was extracted from the proposed ICA. Connectivity of the medial frontal gyrus, superior frontal gyrus and anterior cingulate increased in MS, while the inferior temporal gyrus and superior parietal lobule decreased.

## Figures and Tables

**Figure 1 diagnostics-12-02263-f001:**
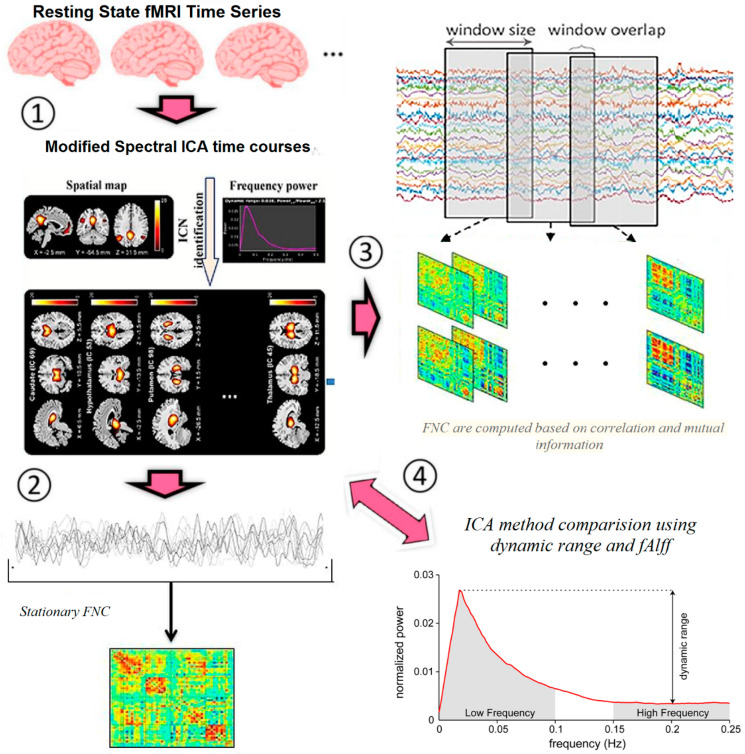
Method block diagram.

**Figure 2 diagnostics-12-02263-f002:**
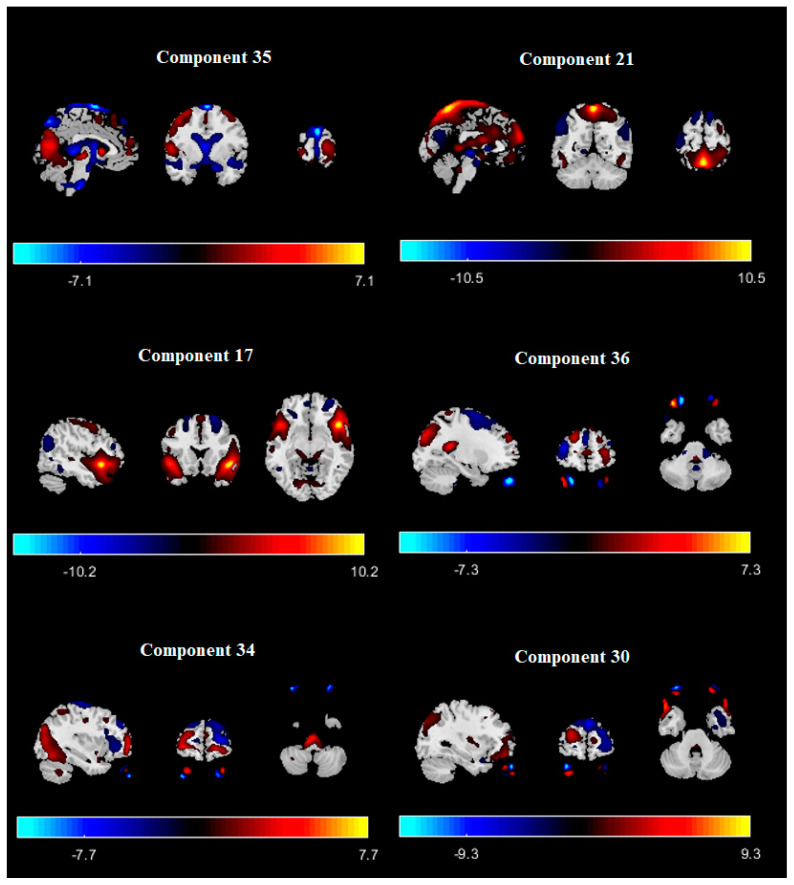
Extracted ICNs with proposed ICA algorithm.

**Figure 3 diagnostics-12-02263-f003:**
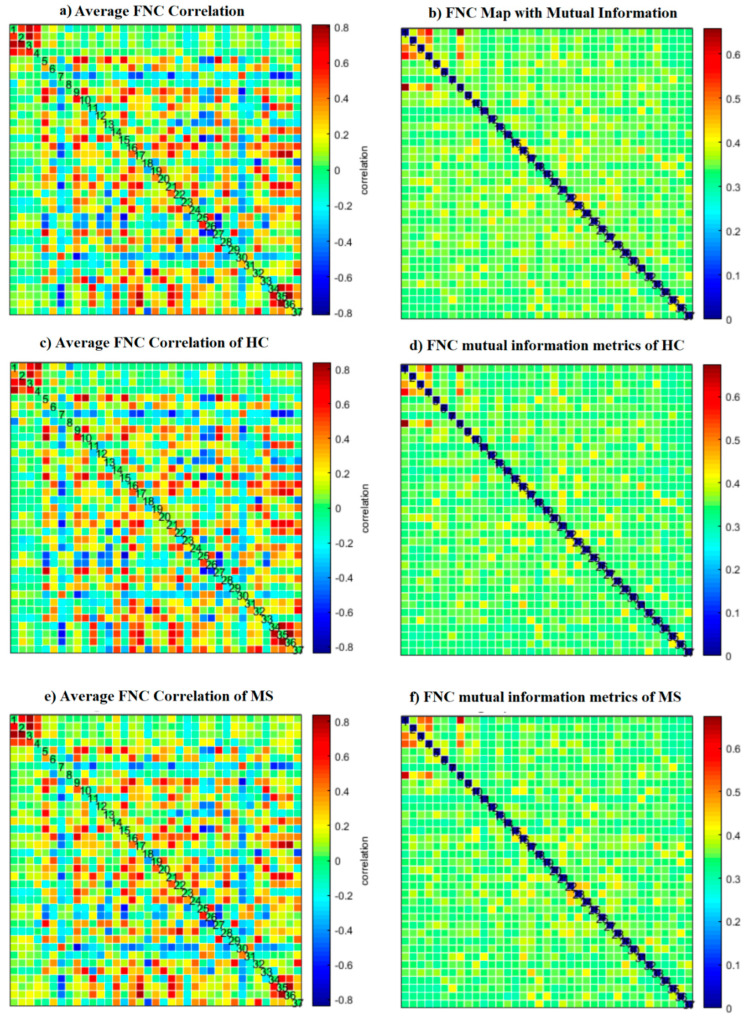
(**a**) Static functional network connectivity correlations were computed for each averaged across sessions. (**b**) FNC metrics of component spatial maps. Mutual information was computed between components and averaged across datasets. (**c**) Functional network connectivity correlations were computed for average HC dataset. (**d**) Mutual information was computed between components spatially for the HC dataset. (**e**) Functional network connectivity correlations we computed for the average MS dataset. (**f**) Mutual information was computed between components spatially for the MS dataset.

**Figure 4 diagnostics-12-02263-f004:**
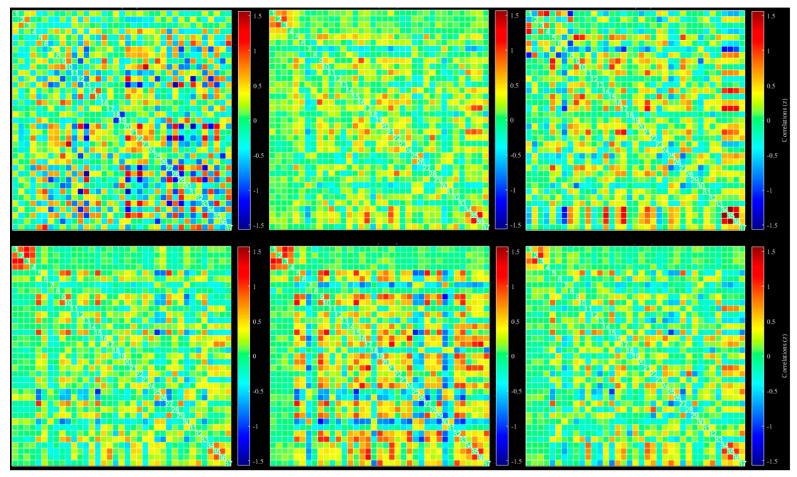
Dynamic functional connectivity of all 37 extracted ICs: six different state, selected based on k-mean clustering.

**Figure 5 diagnostics-12-02263-f005:**
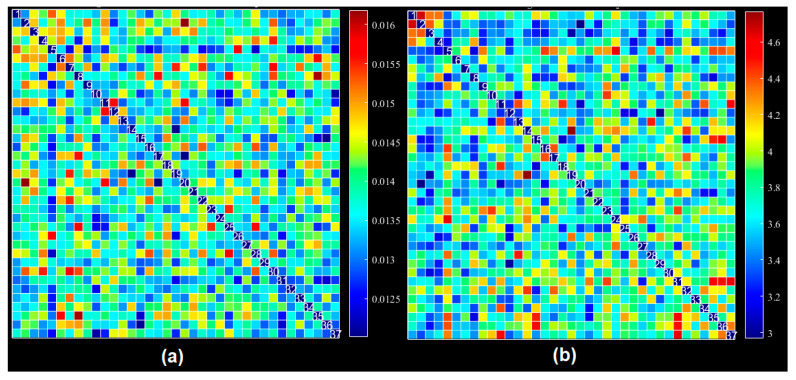
Dynamic functional connectivity using all extracted ICs: (**a**) average FNC across windows, (**b**) average standard deviation across windows for all datasets and averaged across subjects.

**Figure 6 diagnostics-12-02263-f006:**
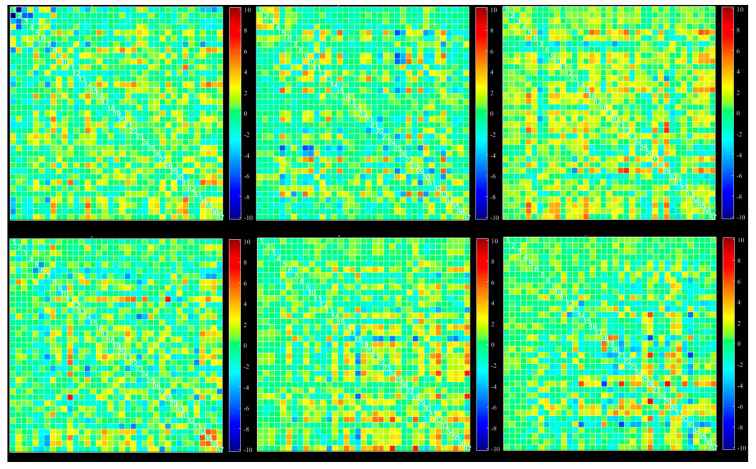
Dynamic functional connectivity of all 37 extracted ICs: six different states, selected based on ICA from all windows.

**Figure 7 diagnostics-12-02263-f007:**
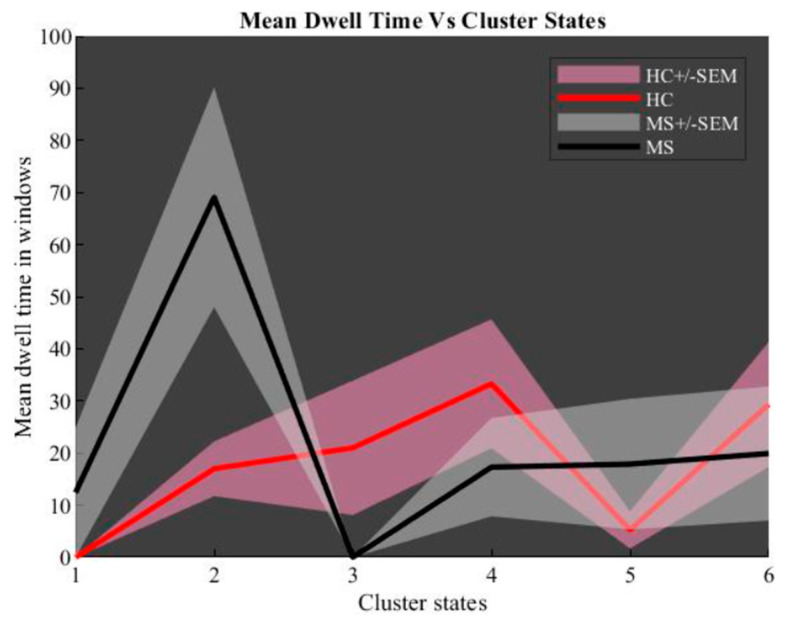
Dynamic connectivity maps: mean dwell time windows vs. cluster states.

**Figure 8 diagnostics-12-02263-f008:**
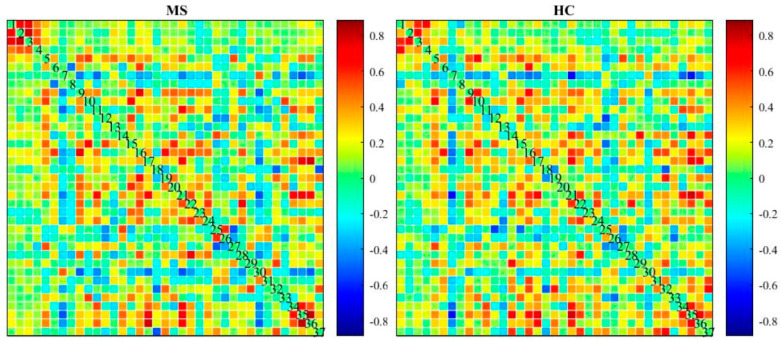
Two-sample t-test for dynamic connectivity map for MS and HC group.

**Figure 9 diagnostics-12-02263-f009:**
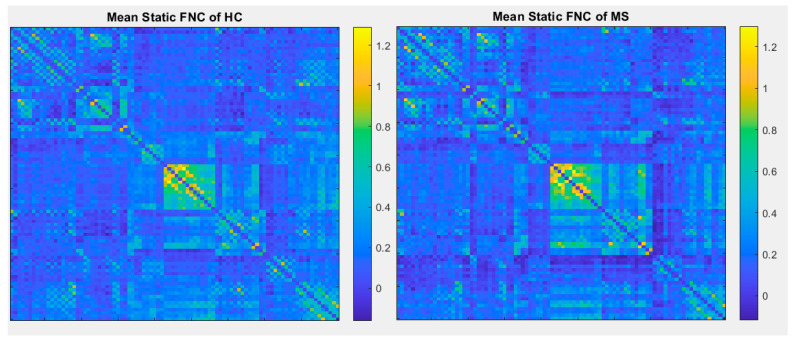
Mean static functional connectivity network for MS and HC using correlation with 90 brain regions defined by AAL atlas.

**Figure 10 diagnostics-12-02263-f010:**
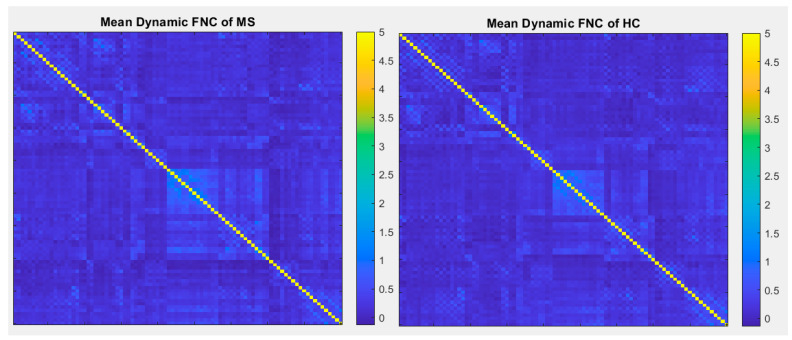
Mean dynamic FNC of both MS group and healthy control group using correlation with 90 brain regions defined by AAL atlas.

**Figure 11 diagnostics-12-02263-f011:**
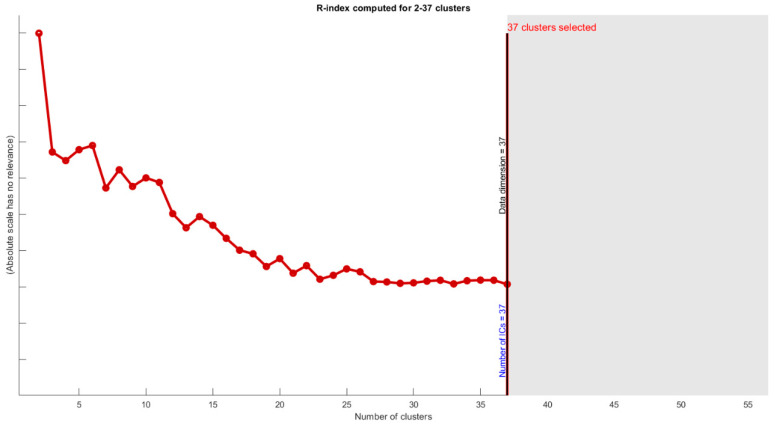
R-index computed for 2–37 cluster showing degree of difference between clusters. Most of the differences will occur for more than 32 clusters.

**Figure 12 diagnostics-12-02263-f012:**
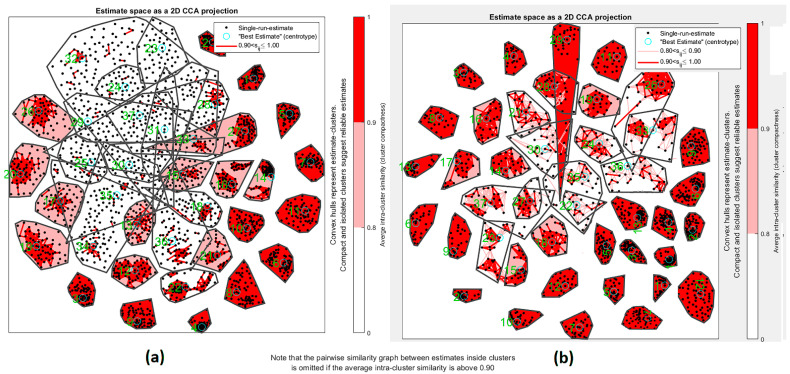
Estimated space as a 2D CCA projection for ICA repetition. Red color shown measure of correlation between clusters. (**a**) Measure of usual ICA based on bootstrapping dataset and repetition of ICA. (**b**) Proposed method of bootstrapping and repetition of ICA.

**Table 1 diagnostics-12-02263-t001:** Studies on MS using fMRI and other MRI modalities.

Group Study	Number of Subjects	Dataset	Method	Follow-Up	Year
RRMS and MS	16 patients with RRMS and 2 patients with possible MS 9 HC	Task fMRI T2-LL T1-LL	Simple motor task	15–26 months	Pantano 2014 [2]
MS	MS (11 cases) HC (9 cases)	rs-fMRI Task fMRI DTI	Cross-correlation Fiber tracking	---	Mark J. Lowe 2008 [3]
RRMS	13 patients with early RRMS 19 HC	rs-fMRI T1, T2	PASAT	12 months	B. Audoin 2008 [4]
RRMS	20 patients with RRMS	rs-fMRI GM volumes and WM architecture	Stroop task and rs-fMRI	12 weeks	M. Filippi 2012 [5]
RRMS	MS (18 cases) HC (18 cases)	rs-fMRI	VBM SEINA ^1^	---	Bonavita 2011 [6]
MS	MS (16 cases) HC (16 cases)	rs-fMRI DTI	SIENA TBSS ^2^	---	Hawellek 2011 [7]
MS	MS (31 cases) HC (31 cases)	rs-fMRI	FEW ^3^	---	Loitfelder 2012 [8]
Early RRMS	MS (13 cases) HC (14 cases)	rs-fMRI	ICA	---	Faivre 2012 [9]
First-stage MS	MS (22 cases) HC (14 cases)	rs-fMRI	Graph connectivity	---	Richiardi 2012 [10]
Different MS	MS (42 cases) HC (42 cases)	Task fMRI	Correlation	6 months	Rocca 2014 [11]
RRMS	MS (55 cases) HC (24 cases)	rs-fMRI	Correlation	12 months	Tona 2014 [12]
MS and NMO	NMO (30 cases) MS (25 cases) HC (35 cases)	rs-fMRI DTI T1, T2	Cross-correlation FA analysis Cortical thickness Analysis	---	Eshaghi 2015 [13]
MS	MS (43 cases) HC (20 cases)	rs-fMRI T1, T2	Correlation Lesion segmentation	---	Zhong 2016 [14]
RRMS and MS	18 cognitively impaired patients with RRMS Control group: 14 cognitively impaired patients with MS	NBV T2-LL rs-fMRI	FSL	8 weeks	S Bonavita 2015 [15]
MS	MS (27 cases) HC (27 cases)	Task fMRI	SPM	---	L Pfaff 2019 [16]
RRMS	MS (12 cases) HC (12 cases)	RS fMRI	CONN toolbox	12 months	V Fleischer 2020 [17]
RRMS and SPMS	40 patients with RRMS and 28 patients with possible SPMS ^4^	RS fMRI	CONN toolbox	---	A Temniy 2021 [18]

^1^ Structural image evaluation using normalization of atrophy; ^2^ Tract-based spatial statistics; ^3^ Family-wise error; ^4^ Secondary progressive MS.

**Table 2 diagnostics-12-02263-t002:** MS and healthy group differences using fMRI.

Brain Areas	BA	Max t-Score	(x, y, z)
MS < HCs			
Inferior Temporal Gyrus	20	4.6	(−57, −36, −21)
Superior Parietal Lobule	7	3.6	(36, −57, 51)
Middle Temporal Gyrus	21, 22	5.1	(63, −45, 3)
Inferior Parietal Lobule	40	3.7	(51, −39, 60)
Lingual Gyrus	18	3.8	(3, −90, −6)
MS > HCs			
Medial Frontal Gyrus	10	17.1	(−6, 63, 6)
Superior Frontal Gyrus	10	8.1	(−12, 69, 3)
Anterior Cingulate	25	3.7	(−3, 18, −3)
Caudate	-	10.4	(−6, 12, 9)
Lateral Ventricle	-	8.6	(−3, 6, 9)
Medial Frontal Gyrus	10, 11	9.8	(−6, 60, −12)
Superior Temporal Gyrus	38	6.7	(−45, 21, −27)
Inferior Temporal Gyrus and Fusiform Gyrus	37	4.3	(−54, −60, −12)
Cuneus	18	4.7	(−9, −96, 15)
Inferior Frontal Gyrus	10, 46	4.5	(−54, 21, 24)

**Table 3 diagnostics-12-02263-t003:** Proposed ICA and Infomax ICA comparison with dynamic range and fALFF of extracted components.

	Proposed ICA		Infomax ICA	
Component Number	Dynamic Range	fALFF	Dynamic Range	fALFF
1	0.038	1.642	0.040	2.133
2	0.033	1.226	0.040	1.540
3	0.033	1.166	0.038	1.926
4	0.029	0.923	0.049	2.541
5	0.040	1.862	0.033	1.235
6	0.033	1.248	0.034	1.241
7	0.034	1.402	0.039	1.857
8	0.034	1.344	0.041	1.953
9	0.038	1.653	0.032	1.166
10	0.035	1.423	0.029	0.958
11	0.039	1.951	0.037	1.368
12	0.030	1.098	0.032	1.073
13	0.038	1.523	0.031	0.993
14	0.039	1.934	0.042	2.033
15	0.036	1.650	0.033	1.247
16	0.040	1.775	0.033	1.251
17	0.039	2.239	0.028	0.824
18	0.036	1.314	0.036	1.333
19	0.038	1.473	0.037	1.748
20	0.033	1.113	0.039	1.948
21	0.040	2.305	0.033	1.204
22	0.037	1.507	0.039	1.536
23	0.038	1.788	0.034	1.298
24	0.039	1.821	0.041	2.146
25	0.036	1.497	0.042	2.341
26	0.034	1.346	0.040	1.955
27	0.036	1.524	0.037	1.207
28	0.041	1.792	0.039	1.914
29	0.038	1.607	0.043	2.217
30	0.045	2.056	0.029	0.869
31	0.039	1.748	0.037	1.632
32	0.036	1.517	0.040	1.726
33	0.039	1.719	0.033	1.265
34	0.041	2.131	0.043	2.347
35	0.043	2.977	0.037	1.738
36	0.042	2.221	0.040	1.972
37	0.038	1.512	0.035	1.536

## Data Availability

Not applicable.

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
