# Peer review of "Dynamic Brain Connectivity in Resting-State FMRI Using Spectral ICA and Graph Approach: Application to Healthy Controls and Multiple Sclerosis"

_diagnostics, 2022, doi:10.3390/diagnostics12092263_

Round 1

Reviewer 1 Report

The manuscript entitled “Dynamic Brain Connectivity in Resting State FMRI Using Spectral ICA and Graph Approach: Application to Healthy Control and Multiple Sclerosis” has been investigated in detail. The topic addressed in the manuscript is potentially interesting and the manuscript contains some practical meanings, however, there are some issues which should be addressed by the authors:

1)      In the first place, I would encourage the authors to extend the abstract more with the key results. As it is, the abstract is a little thin and does not quite convey the interesting results that follow in the main paper. The "Abstract" section can be made much more impressive by highlighting your contributions. The contribution of the study should be explained simply and clearly.

2)      The readability and presentation of the study should be further improved. The paper suffers from language problems.

3)      The importance of the design carried out in this manuscript can be explained better than other important studies published in this field. I recommend the authors to review other recently developed works.

4)      What makes the proposed method suitable for this unique task? What new development to the proposed method have the authors added (compared to the existing approaches)? These points should be clarified.

5)      “Discussion” section should be edited in a more highlighting, argumentative way. The author should analysis the reason why the tested results is achieved.

6)      The effect of the parametric uncertainty is not discussed in detail. How did the comparison methods perform with or without the uncertainty?

7)      It will be helpful to the readers if some discussions about insight of the main results are added as Remarks.

This study may be proposed for publication if it is addressed in the specified problems.

Author Response

REPLY TO REVIEWER 1’s COMMENTS:

We appreciate your valuable comments suggesting how to improve our paper. All answers to Reviewer 1’s Comments (in the paper) are highlighted in yellow color.

Line Number for Correction

Comment (1):

In the first place, I would encourage the authors to extend the abstract more with the key results. As it is, the abstract is a little thin and does not quite convey the interesting results that follow in the main paper. The "Abstract" section can be made much more impressive by highlighting your contributions. The contribution of the study should be explained simply and clearly.

1

Reply:

Thanks to your comment, the new abstract is now prepared.

Comment (2):

The readability and presentation of the study should be further improved. The paper suffers from language problems.

Reply:

It is now improved.

Comment (3):

The importance of the design carried out in this manuscript can be explained better than other important studies published in this field. I recommend the authors to review other recently developed works.

353-360

369-384

Reply:

We addressed the comparison and importance of this method in discussion.

Comment (4)- Minor:

What makes the proposed method suitable for this unique task? What new development to the proposed method have the authors added (compared to the existing approaches)? These points should be clarified.

Reply:

In fact, MS is characterized by a particularly widespread and severe damage mainly affecting the white matter that can cause FC alterations secondary to structural disconnection between RSN nodes.

RSNs abnormalities have been found in almost all multiple sclerosis (MS) phenotypes.

There are lots of ICA method like infomax, fast ICA and etc for blind source seperation. Also, there are lots of differenet implementation in ICA like RELICA, JADE and etc. Proposed method uses spectral ICA that is proved mathemathicly. All these methods are able to use for these task but in these paper we use ICA method based on Infomax algorithm and ICA method based on proposed method and compare their results. Proposed method could be use in all other tasks that blind source seperation is needed. Comparison is done in Table 2

The aim of the fMRI application is to detect different RSNs and to investigate their involvement in specific functions. The two most commonly applied methods for RS investigation are the region-of-interest (ROI) analysis and the whole brain investigation, the latter consisting mainly of the ICA. The ROI analysis correlates the time course of a predefined ROI with other brain voxels, according to the detection of coherent BOLD fluctuations. However, this approach is limited by the relative arbitrariness of the ROI selection. Conversely, ICA is a data-driven, whole-brain approach, designed to separate a multivariant signal in its sub-components, thus providing a single signal from a complex of signals. ICA is used without any a priori hypothesis and assuming the statistical independence of the sources and the BOLD signal is decomposed into spatially and temporally distinct maps with their own time courses. Each map may be interpreted as a network of brain regions that share similar BOLD fluctuations over time.

Comment (5)- Minor:

“Discussion” section should be edited in a more highlighting, argumentative way. The author should analysis the reason why the tested results is achieved.

305-311

Reply:

Thanks for comment. Discussion is now changed and we added types of comparisons with highlighted changes.

Comment (6)- Minor:

The effect of the parametric uncertainty is not discussed in detail. How did the comparison methods perform with or without the uncertainty?

279-300

Reply:

Considering two types of uncertainties in ICA(number of ICs and repeating ICA) we added our strategy  to solve the uncertainty in this paper.

Comment (7)- Minor:

It will be helpful to the readers if some discussions about insight of the main results are added as Remarks.

374-380

Reply:

Discussions about insight of the main results is added.

Reviewer 2 Report

  1. 1. Line 97, the number of the document (ethical statements) should be presented here.
  2. 2. Methods. In parts C and D, all conclusions and reasons to choose that or another method should be moved to the results and discussions. The section Methods should include brief description on how the method was used, what is taken into account, how many measurements were provided, what was used for this and so on. See for example lines 174-187, the starting text is not method description.
  3. 3. Lines 286-287, is should be replaced by are.

Author Response

REPLY TO REVIEWER 2’s COMMENTS:

We appreciate your valuable comments suggesting how to improve our paper. All answers to Reviewer 2’s Comments (in the paper) are highlighted in green color.

Line Number for Correction

Comment (1):

Line 97, the number of the document (ethical statements) should be presented here.

102-104

Reply:

Thanks for the comment, we added this to the manuscript..

Comment (2):

Methods. In parts C and D, all conclusions and reasons to choose that or another method should be moved to the results and discussions. The section Methods should include brief description on how the method was used, what is taken into account, how many measurements were provided, what was used for this and so on. See for example lines 174-187, the starting text is not method description.

Reply:

Thanks. We applied the proposed comments.

Comment (3):

Lines 286-287, is should be replaced by are.

Reply:

Done.

Round 2

Reviewer 1 Report

All my comments have been thoroughly addressed. It is acceptable in the present form.